# Simplification with Boosted Protease Inhibitor-Based ART Versus Maintenance of Conventional ART: Results from a Five-Year Controlled Cohort

**DOI:** 10.3390/v17060751

**Published:** 2025-05-24

**Authors:** Mateus Swarovsky Helfer, Guilherme Carvalho Serena, Tarsila Vieceli, Eduardo Sprinz

**Affiliations:** 1Infectious Disease Department, Hospital de Clínicas de Porto Alegre, Rio Grande do Sul 90035-903, Brazil; mateuswahelfer@gmail.com; 2Postgraduate Program in Medical Sciences, Federal University of Rio Grande do Sul, Porto Alegre 90035-903, Brazil; 3School of Medicine, Federal University of Rio Grande do Sul, Porto Alegre 90035-903, Brazil; guilhermeserena@gmail.com; 4Infectious Diseases Department, Santa Casa de Porto Alegre, Porto Alegre 90020-090, Brazil; tarsilavieceli@gmail.com

**Keywords:** AIDS, HIV, simplification therapy, boosted protease inhibitor, integrase inhibitor sparing regimen

## Abstract

Dolutegravir-based antiretroviral therapy (ART) simplification is increasingly common, although some patients cannot take this drug due to intolerance or drug resistance. Boosted-protease inhibitors (bPI) might be an option in this scenario. Nevertheless, long-term outcomes have not been studied yet. A controlled cohort study comparing 5-year outcomes of ART simplification bPI-based regimens (without integrase strand transfer inhibitor—INSTI) versus ART maintenance was conducted in a Brazilian referral center. Viral suppression rates and mortality after 5 years were the primary outcomes of the study. Eighty individuals were included in each group; 47.5% were women, and the mean age was 56 years. The five-year survival rate was 88.8% in the simplified group and 87.5% in the maintenance arm (log-rank = 0.41). Viral suppression rate was 78.8% and 70.0%, respectively (*p* = 0.28). Individuals presented less renal function decline (−5 vs. −10 mL/min/1.73 m^2^; *p* < 0.05) in the simplified arm. No difference was observed in metabolic parameters. Based on our findings, ART simplification without INSTI has shown efficacy and safety comparable to maintenance of triple therapy even in the long term, and could be an option in these situations, which might be even more important in settings with limited options.

## 1. Introduction

Highly active antiretroviral therapy (HAART) dramatically improved the life expectancy of people living with HIV (PLWHIV) [1]. However, as this population aged, an increased prevalence of non-AIDS-related conditions, such as cardiovascular and metabolic diseases, became more commonly diagnosed [2]. These comorbidities could be associated with HIV disease and its complications, the accelerated aging process related to chronic inflammation due to HIV infection, and antiretroviral drugs [3]. Antiretroviral drug therapy (ART) simplification has been tried with the objective to mitigate problems related to drug toxicity and adherence to treatment since the end of the last century; nevertheless, the initial trials had shown that this approach was inferior to maintenance of HAART regarding maintaining HIV viral suppression [4,5].

Newer antiretrovirals, with better pharmacokinetic profiles, more potency, and less toxicity, improved some gaps related to therapy [6]. The development of potent antiretrovirals, such as the latest boosted protease inhibitors (bPI) and second-generation integrase strand transfer inhibitors (INSTI), leads to a higher antiretroviral potency and genetic barrier [6,7]. Again, ARV simplification has gained a new window [8,9,10]. This approach possibly would reduce treatment-related toxicity and improve the excess burden of chronic comorbidities in PLWHIV.

ART simplification, mainly with dolutegravir (DTG) plus lamivudine (3TC), has become widespread in recent years, and it is indicated as an option in several HIV clinical guidelines [11,12]. Other ARV combinations recommended in these guidelines for simplified regimens include boosted darunavir (bDRV) combined with either 3TC or NNRTI.

However, several knowledge gaps are still desirable to be addressed regarding ARV simplification. First, clinical studies have used only surrogate markers, such as viral suppression, to assess primary outcomes so far [13,14]. Therefore, developing more data about the impact on hard outcomes, such as mortality, is essential. Secondly, the short follow-up period of major studies—usually limited to 48 to 96 weeks—leaves a gap in our still understanding long-term outcomes of this approach.

Another important aspect is that most trials focused only on DTG-based regimens, which could be a limitation. Not every individual is going to be a suitable candidate for this drug. The extensive use of INSTI in first-line therapy creates an unfavorable environment with the possibility for the increased prevalence of INSTI-associated drug resistance mutations. Some recent reports [15,16] indicated a resistance prevalence of around 6% in first-line and up to 16–22% in second-line INSTI therapy failures. Additionally, the use of DTG has been associated with excessive weight gain in certain specific populations, such as Black women [17], and issues related to intolerance and adverse effects, which may affect up to 13% of patients, would further limit the use of DTG [18]. As a result, it is important to develop and evaluate alternative ART simplification regimens, particularly in low-income countries, where antiretroviral options are limited and overdependence on a single class might be a concern. This study aimed to fill some gaps regarding ARV simplification, such as studying boosted PI regimens as the core of the simplified regimen and for a longer period of time (at least 5 years).

## 2. Materials and Methods

### 2.1. Study Design

A retrospective cohort study comparing 5-year outcomes after ART simplification versus ART maintenance was conducted at the HIV outpatient clinic of Hospital de Clínicas de Porto Alegre, a national HIV referral center in the southernmost state of Brazil. The study was approved by the ethics research committee (CAEE no. 65271922.4.0000.5327).

### 2.2. Participants

Individuals who had (a) an undetectable HIV viral load for at least one year and (b) had their antiretroviral regimen switched to a two-drug regimen without integrase inhibitors were consecutively enrolled in the ART simplification group (referred to as the simplification group) through a medical record review from January 2004 through December 2018. Individuals in the ART maintenance group (referred to as the control group), matched by sex and age, also had undetectable HIV viral loads for at least one year and were selected on the same day of clinic attendance. If no suitable control could be identified on a given day, the search would go on to the closest day. Exclusion criteria was active hepatitis B (HBsAg-positive individuals).

### 2.3. Procedures

The 5-year follow-up period began on the day of treatment simplification. Individuals were monitored through clinical visits and laboratory tests at least every six months. Data was collected through medical record review. We performed a survival analysis using an intention-to-treat (ITT) approach, which included all subjects in the group, and a per-protocol (PP) approach, which included all ITT patients except those who discontinued therapy due to adverse effects. HIV viral load, lipid profile, and renal laboratory data were evaluated at the baseline and 5 years. Regimen simplification was defined as a change to a dual regimen in an individual with a suppressed HIV viral load for at least one year. ART modification was defined by an individual in the simplification group who returned to triple therapy or an individual in the control group who had his ART regimen simplified. Virological failure was defined by two consecutive measurements of HIV viral load of ≥50 copies/mL.

### 2.4. Outcomes

The primary outcomes were all-cause mortality and virological suppression rate after five years. Secondary outcomes were metabolic and renal parameters and ART modification (simplification group individuals back to triple therapy and control group individuals’ simplification).

### 2.5. Statistical Analysis

Categorical data were analyzed using chi-square or Fisher’s exact tests, as appropriate. Continuous variables were compared using the Mann–Whitney U test. Survival analysis was performed using Kaplan–Meier curves and the log-rank test. Glomerular filtration rate (eGFR) was calculated by the CKD-EPI equation. Missing data were censored. The analyses were performed using IBM SPSS Statistics software version 27.

### 2.6. Sample Size Calculation

Historical data from our center indicated that patients on antiretroviral therapy had a 5% incidence of mild renal function decline per year, which doubled when they were exposed to tenofovir disoproxil fumarate (TDF) [19]. Therefore, we estimated that patients on triple therapy would have an annual incidence decline of 10% in renal function, compared to 5% in patients on dual therapy (without TDF). Over five years, the cumulative incidence would be 50% in patients on triple therapy and 25% in dual therapy patients. These values were used as parameters for the sample size calculation. Sixty-four patients would be required in each group to achieve 80% power. However, we set the target at 80 patients per group to account for potential losses (25% more participants).

## 3. Results

### 3.1. Baseline Characteristics

Eighty individuals were included in each group (Table 1). Mean age was 56.8 years; 47.5% were women, 84.4% self-declared as white, and 12.5% as Black. The majority of individuals (86.3%) met the defining AIDS CDC criteria [20]. At the beginning of the study follow-up (baseline), the median CD4 count of the individuals was 526 cells/mm^3^ (median nadir CD4 count was 101 cells/mm^3^). Individuals in the simplification group had a longer duration of HIV infection (4.9 years; *p* = 0.022) and longer exposure to antiretroviral therapy (4.3 years; *p* = 0.015).

The most frequent comorbidities at baseline were dyslipidemia (54.4%), hypertension (41.3%), chronic kidney disease (20.6%), and diabetes mellitus (19.4%). The simplification group showed higher rates of hypertension (53.8% vs. 28.7%; *p* = 0.002) and chronic kidney disease (33.8% vs. 7.5%; *p* = 0.001). The simplification group also had lower eGFR and higher triglyceride levels. A full description of baseline characteristics is presented in Table 1.

The main reasons for ART simplification were adverse events (96.2%) and dosing convenience (3.8%). The most prescribed double combinations were ATV/r+3TC (57.5%), followed by ATV/r+EFZ (17.5%) and LPV/r+3TC (10.0%). A full description of antiretroviral regimens is presented in Appendix A Table A1.

### 3.2. Virological Suppression

A total of 119 participants (74.4%) had viral suppression at the end of the 5-year follow-up (ITT analysis), including 63 (78.8%) in the simplification group and 56 (70.0%) in the control group (*p* = 0.208). There were 3 virological failures in the simplification group and 2 in the control group (*p* = 0.650). No mutation associated with resistance was detected in the cases of viral failure.

### 3.3. Survival

The overall 5-year survival was 88.1% in the ITT analysis. There were 18 deaths during the five-year follow-up (7 in the simplification group and 11 in the control group). There were no deaths attributable to opportunistic diseases or AIDS; however, information about the cause of death could not be assessed in a significant proportion of cases (*n* = 11; 61.1%). The determined causes of death included cirrhosis (*n* = 2), hematologic malignancy (*n* = 2), Parkinson’s disease, renal cancer, and heart failure (1 each).

In the ITT analysis, the five-year survival rate was 88.8% in the simplification group vs. 87.5% in the control group (log-rank = 0.413). Regarding PP analysis, the five-year survival rate in the simplification group was 90.7%, and 86.9% in the control group (log-rank = 0.121). Kaplan–Meier tables are presented in Figure 1.

There was no difference between the groups regarding the other causes of treatment failure over the five years (the most frequent causes being death, 7 in simplification vs. 11 in control group; and discontinuation due to adverse effects, 7 in simplification vs. 8 in control group. A complete description of the five-year outcomes is provided in Table 2.

### 3.4. Lipid and Renal Assessments

Lipids remained stable over the five years, with no significant difference between groups. There was no difference in serum creatinine; however, there was a lower reduction in eGFR over the five years in the simplification group (decrease of 5 mL/min/1.73 m^2^ compared to 10 mL/min/1.73 m^2^ in the control group; *p* = 0.009). A full description of lipid and renal can be assessed in Table 3.

## 4. Discussion

This is a real-world observational study conducted on an extensively exposed population to ART (median ART duration of 10 years). To our knowledge, this is the first study to address long-term outcomes (over five years) of simplification therapy without INSTI, as there are only a few studies and only with the 3TC+DTG combination that addressed outcomes beyond 144 weeks [21,22]. In this cohort studied, simplification with bPI regimens, without INSTI, was not associated with a higher risk of virological failure or death when compared to triple therapy maintenance.

Despite age matching, the simplification group exhibited a higher prevalence of non-AIDS-related comorbidities. This finding may be explained by the longer duration of HIV infection in this group, which increases the exposure to factors like chronic inflammation linked to HIV and (likely) more prolonged exposure to antiretrovirals (known for their metabolic toxicity, such as first- and second-generation ones) [23]. This higher burden of chronic comorbidities could be expected to be associated with poorer outcomes (including increased mortality). However, mortality in the simplification group was not higher, suggesting that factors beyond comorbidities may influence survival outcomes in this population (such as improved management of comorbidities and advances in antiretroviral therapy).

The longer exposure to antiretroviral therapy in the simplified group (4.3 years) is another aspect that could result in worse outcomes regarding viral failure. The chance of having been exposed to weaker ART regimens and, consequently, a higher chance of drug resistance could have happened in this group. Such exposure increases the likelihood of the occurrence of archived resistance mutations, which are often associated with higher risks of virological failure [24]. However, there was no significant difference in virological failure between the groups, indicating that these regimens remain effective despite concerns regarding previous treatment failures.

Although group simplification exhibited a higher prevalence of hypertension and lower eGFR at baseline, this group demonstrated less renal function decline, showing a decrease of 7% (−5 of 68 mL/min/1.73 m^2^) compared to 11% (−10 of 93 mL/min/1.73 m^2^) in the control group (*p* = 0.009). Therefore, we can speculate that treatment simplification could have played a role in nephroprotection, as previously observed in some simplification studies [25,26]. Nephroprotection measures are crucial for this population, as they may be at jeopardy for the development of chronic kidney disease according to the risk score developed by the D:A:D study [27], particularly regarding age and the prevalence of comorbidities like hypertension and diabetes mellitus.

When comparing the data from our cohort with studies with 3TC+DTG simplification, it is essential to refer to real-world studies with long-term follow-up. Maggiolo et al. [21] followed 218 patients for 5 years in a single-arm cohort. There was no viral failure, and 77% of patients remained on the regimen till the end of the study (discontinuations were primarily due to death or adverse events). Similarly, Ciccullo et al. [22] also reported a cohort study without a comparator group of 518 patients, in which 74% of patients continued on the regimen after 5 years. There were 18 cases of viral failure (3.5%) and an 11% discontinuation rate due to toxicity. In our study, likewise, 78.8% of patients remained on therapy. There was 3.8% of virological failure and 8.8% discontinuations due to adverse effects, which also could be comparable to these studies. The frequency of discontinuations due to adverse events in our cohort was similar to those reported in real-world studies involving 3TC+DTG.

The study has some limitations inherent to observational methodology. Despite efforts to match groups, baseline differences persisted, like the higher prevalence of some comorbidities, longer duration of HIV infection, and longer exposure to ART. Another key limitation is the incomplete information on causes of death, with 61% of cases lacking accessible data. However, these patients had confirmed viral suppression in the months preceding death, making AIDS-related mortality unlikely. Another limitation is that the milder decline of eGFR in the simplification group may reflect channeling bias, as treatment simplification is often prescribed for patients with impaired renal function. In this way, the greater decline in glomerular filtration rate among patients on triple therapy may be attributed to their higher baseline eGFR (regression to mean phenomenon).

In conclusion, our cohort data are compelling. The findings of our study support that bPI simplification regimens (without INSTI) were comparable to standard ART maintenance. Long-term dual therapy with protease inhibitors combined with 3TC or NNRTIs was not associated with higher mortality nor higher risk of virological failure. Newer antiretroviral dual combinations also should be explored in the near future (e.g., 3TC plus bDRV or islatravir plus doravirine). Therefore, this approach, even without INSTIs, could be a viable strategy when dolutegravir may not be an option, especially in constrained settings with limited ART options.

## Figures and Tables

**Figure 1 viruses-17-00751-f001:**
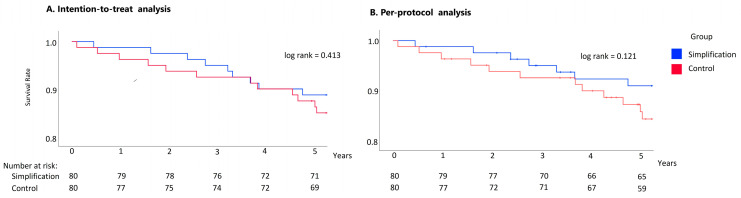
Kaplan–Meier survival curves: (**A**) intention-to-treat and (**B**) per-protocol analysis.

**Table 1 viruses-17-00751-t001:** Baseline characteristics.

	Simplification (80)	Control (80)	*p*
Age, years (mean ± SD)	57.2 ± 11.2	56.5 ± 10.7	0.692
Sex at birth, female—*n* (%)	38 (47.5)	38 (47.5)	0.999
Self-declared ethnicity—*n* (%)			0.816
Afrodescendent	11 (13.8)	9 (11.3)	
Caucasian	67 (83.8)	68 (85.0)	
CD4 nadir (cells/dL)	102 (43–218)	91 (32–384)	0.616
AIDS—*n* (%)	71 (88.8)	67 (83.8)	0.491
Time since HIV diagnosis (years)	17.1 (12.8–20.7)	12.2 (6.7–19.5)	0.022
Reason for ART initiation—*n* (%)			0.333
Opportunistic disease	23 (44.2)	20 (34.5)	
CD4 < 200 (cells/dL)	23 (44.2)	22 (37.9)	
CD4 < 350 (cells/dL)	3 (5.8)	8 (13.8)	
Universal ART	4 (7.7)	8 (13.8)	
Time since ART initiation (years)	13.0 (7.4–17.1)	8.7 (3.9–14.8)	0.015
Baseline CD4 (cells/dL)	558 (358–869)	510 (377–682)	0.377
NRTI at the baseline			
Tenofovir (TDF)	36 (45.0)	35 (43.8)	0.999
Zidovudine (AZT)	25 (31.3)	38 (60.3)	0.052
Abacavir (ABC)	9 (11.3)	6 (7.5)	0.588
NRTI at the baseline			
Tenofovir (TDF)	36 (45.0)	35 (43.8)	0.999
Zidovudine (AZT)	25 (31.3)	38 (60.3)	0.052
Abacavir (ABC)	9 (11.3)	6 (7.5)	0.588
Comorbidities—*n* (%)			
Hypertension	43 (53.8)	23 (28.7)	0.002
Diabetes	18 (22.5)	13 (16.3)	0.424
Dyslipidemia	49 (61.3)	38 (47.5)	0.112
Osteoporosis	17 (21.3)	12 (15.0)	0.412
Ischemic heart disease	4 (5.0)	2 (2.5)	0.677
Other arterial disease	5 (6.3)	6 (7.5)	0.999
CKD	27 (33.8)	6 (7.5)	0.001
HAND	2 (2.5)	4 (5.0)	0.677
Hepatitis C	7 (8.8)	6 (7.5)	0.999
Creatinine (mg/dL)	1.00 (0.79–1.26)	0.80 (0.70–0.92)	0.001
eGFR (mL/min/1.73 m)	68 (52–91)	93 (78–105)	0.001
Total cholesterol (mg/dL)	185 (156–218)	184 (157–211)	0.469
HDL (mg/dL)	43 (36–49)	45 (38–54)	0.205
LDL (mg/dL)	101 (79–133)	104 (89–125)	0.859
Triglycerides (mg/dL)	191 (124–253)	146 (92–207)	0.006

Abbreviations: ART, antiretroviral therapy; CKD, chronic kidney disease; HAND, HIV neurocognitive associated disorder; eGFR, estimated glomerular filtration rate.

**Table 2 viruses-17-00751-t002:** Comparison of the primary outcome after five years of follow-up (intention to treat analysis).

Five-Year Outcome	Overall, 160 (%)	Simplification, 80 (%)	Control, 80 (%)	*p*
Suppressed viral load	119 (74.4)	63 (78.8)	56 (70.0)	0.277
Dead	18 (11.3)	7 (8.8)	11 (13.8)	0.454
Discontinuation due to adverse effects	15 (9.4)	7 (8.8)	8 (10.0)	0.786
Loss of follow-up	3 (1.9)	0 (0.0)	3 (3.8)	0.245
Viral load ≥ 50 copies/mL or discontinuation due to failure	5 (3.1)	3 (3.8)	2 (2.0)	0.650

**Table 3 viruses-17-00751-t003:** Comparison of the metabolic and renal parameters variation after five years of follow-up.

Five-Year Variation	Simplification	Control	*p*
Total cholesterol (mg/dL)	0 (−43; 37)	−9 (−32; 21)	0.520
HDL (mg/dL)	−1 (−5; 5)	0 (−7; 7)	0.768
LDL (mg/dL)	7 (−28; 46)	−6 (−26; 27)	0.417
Triglycerides (mg/dL)	−23 (−79; 29)	−13 (−64; 23)	0.967
Creatinine (mg/dL)	0.07 (−0.10; 0.29)	0.10 (−0.20; 0.24)	0.423
eGFR (mL/min/1.73 m^2^)	−5 (−14; 5)	−10 (−22; −3)	0.009

## Data Availability

Supporting data are available from the corresponding author upon request.

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
