# Peer review of "Simplification with Boosted Protease Inhibitor-Based ART Versus Maintenance of Conventional ART: Results from a Five-Year Controlled Cohort"

_viruses, 2025, doi:10.3390/v17060751_

Round 1
Reviewer 1 Report
Comments and Suggestions for Authors
This manuscript provides information on ART simplification to bPI regimens.
Only minor
The main advantage of this paper is the long follow up of patients receiving simplified bPI regimen compared to those who maintained to previous ART.Nevertheless I have some concerns regarding this ms.
First of all simplification arm received PIs that are no longer in most guidelines. ATV/r LPV/r are no longer included in European or American guidelines. There is no data regarding Darunavir which is the only PI that is used (mentioned in line 51)
In the abstract lines 13-14: it needs to be clarified that long term outcomes are not available for simplified bPI regimens. Similarly, I would suggest that the title should start with the word "Simplification".
I believe that HBV co-infection was an exclusion criteria(for both groups). If yes, it should be emphasized
No history of mutations is available. This is a major limitation of the study. In both arms it is essential to have this information, whether it was available or not.
Above all, the major limitation of the study is that there is lack of data on cause of death. The primary endpoint was mortality rate and yet there is no information in a significant proportion of cases.
Author Response
We sincerely thank the reviewer for their valuable comments and suggestions, which have contributed to improving the clarity and quality of our manuscript.
Comments 1:
First of all, the simplification arm received PIs that are no longer in most guidelines. ATV/r LPV/r are no longer included in European or American guidelines. There is no data regarding Darunavir which is the only PI that is used (mentioned in line 51)
Response 1:
The fact that darunavir was present in only 5% of the simplified regimens (supplementary table A1) reflects the limited availability of this antiretroviral in our country during the study period. Additionally, we acknowledge that most of the PIs used (ATV/r and LPV/r) are no longer recommended in current guidelines (also in ours nowadays), but were widely used in our country at the time the study was initiated.
Comments 2:
In the abstract lines 13-14: it needs to be clarified that long term outcomes are not available for simplified bPI regimens. Similarly, I would suggest that the title should start with the word "Simplification".
Response 2
Thank you for the suggestion. We tried to address this issue and we also changed the title.
Comments 3:
I believe that HBV co-infection was an exclusion criteria(for both groups). If yes, it should be emphasized.
Response 3:
Yes, hepatitis B infection was an exclusion criteria, and we have added this information to the Methods section.
Comments 4:
No history of mutations is available. This is a major limitation of the study. In both arms it is essential to have this information, whether it was available or not.
Response 4:
Indeed, no data on baseline mutation history were available. It is important to note that all individuals had suppressed viral load for at least 1 year before being enrolled in the trial.
Comments 5:
Above all, the major limitation of the study is that there is lack of data on cause of death. The primary endpoint was mortality rate and yet there is no information in a significant proportion of cases.
Response 5:
We fully agree with your observation, as written in the discussion in the limitations paragraph. This is a real world data and also can be related to its retrospective and observational nature, which made it impossible to retrieve information in cases of deaths occurring outside of our institution.
Reviewer 2 Report
Comments and Suggestions for Authors
This study tested the effect of boosted PI-based, simplified ART without INSTI in comparison to conventional ART. The authors recruited 80 ARTs and 80 ARTm individuals, and a long-term follow-up for five years was carried out. The two groups showed no statistically significant differences in the 5-year survival rates. On the other hand, the ARTs group exhibited poorer renal function than the ARTm group, possibly due to the longer duration of treatment. However, the results suggest that the simplified regimen could be protective for renal function. The study results are interesting. I want to comment on a couple of issues that need to be addressed.
It should be confirmed whether the observed gap in renal function between the two groups was really derived from the difference in the duration of treatment. For that purpose, subsets of ARTs and ARTm to match “Time since HIV diagnosis” can be compared.
In Tables, the ARTs group is described as “Dual Therapy” whereas the ARTm group is defined as “Control”. The groups' names can be integrated into ARTs/ARTm or Dual Therapy/Control. The meanings of ARTs and ARTm can be better summarized to indicate which two or three drugs were included in the regimens.
Author Response
We sincerely thank the reviewer for their valuable comments and suggestions, which have contributed to improving the clarity and quality of our manuscript.
Comments 1:
It should be confirmed whether the observed gap in renal function between the two groups was really derived from the difference in the duration of treatment. For that purpose, subsets of ARTs and ARTm to match “Time since HIV diagnosis” can be compared.
Response 1:
It is important to emphasize that ARTs (now simplification group) at baseline had significantly more years exposed to ARV than ARTm (now control group), and in the 5-year period of observation of the study there was little bit higher loss of renal function in the ARTm group, this could influenced by the use of some nephrotoxic agents and/or regression to means phenomenon.
Comments 2.
In Tables, the ARTs group is described as “Dual Therapy” whereas the ARTm group is defined as “Control”. The groups' names can be integrated into ARTs/ARTm or Dual Therapy/Control. The meanings of ARTs and ARTm can be better summarized to indicate which two or three drugs were included in the regimens.
Response 2:
Thank you for the observation. We tried to unify the description of groups to Simplification/Control.
Reviewer 3 Report
Comments and Suggestions for Authors
The current manuscript is an important and pragmatic study on ART simplification using bPI regimens in patients who cannot tolerate dolutegravir. Suggestions for improvement:
-
Describe how the simplified and maintenance groups were selected. Were they matched for baseline characteristics (e.g., age, comorbidities, treatment duration)?
-
Indicate inclusion/exclusion criteria, especially regarding prior ART history and reasons for dolutegravir intolerance.
-
Clearly define viral suppression (e.g., <50 copies/mL), and explain what constitutes survival (e.g., all-cause mortality).
-
Clarify how renal function was measured (e.g., eGFR via CKD-EPI formula) and what time points were used.
-
Mention statistical methods for survival analysis (e.g., Kaplan–Meier, Cox proportional hazards), adjustments for confounders, and handling of missing data.
-
Clarify whether intention-to-treat or per-protocol analysis was used.
-
Describe the follow-up protocol (e.g., how often patients were monitored, clinical visits, lab tests).
-
Include any baseline and follow-up adherence measures if available.
- Situate the findings in the context of current WHO or national ART guidelines, especially regarding simplification strategies and INSTI-sparing regimens.
- Discuss potential reasons for less renal function decline in the simplified arm—e.g., removal of tenofovir or dolutegravir, known for nephrotoxic potential in some patients.
-
Address the risk of resistance development under bPI-based simplification, especially in settings with limited genotypic resistance testing.
-
Mention if genotyping was performed pre-simplification.
-
Acknowledge potential biases (e.g., non-randomized design), small sample size, retrospective components, or missing data.
-
Note whether cause of death was analyzed or available.
- Suggest inclusion of newer-generation bPIs or dual regimens (e.g., boosted darunavir + lamivudine) for exploration.
It is recommended that the authors revise the overall English editing and grammar.
Author Response
We sincerely thank the reviewer for their valuable comments and suggestions, which have contributed to improving the clarity and quality of our manuscript.
Comments 1.
Describe how the simplified and maintenance groups were selected. Were they matched for baseline characteristics (e.g., age, comorbidities, treatment duration)?
Response 1.
The simplification group was identified through a screening of follow-up visit records at our tertiary center between 2004 and 2018, according to the criteria described in section “2.2 Participants.” The selection and matching criteria for the controls (age and sex) are also described in the same section.
Comments 2.
Indicate inclusion/exclusion criteria, especially regarding prior ART history and reasons for dolutegravir intolerance.
Response 2.
Inclusion criteria was described in “2.2 Participants”, the only exclusion criteria was HBV infection. Regarding intolerance to DTG, it is important to clarify that the participants were not exposed to DTG. However, our findings provide a rationale for ART simplification in patients with DTG intolerance.
Comments 3.
Clearly define viral suppression (e.g., <50 copies/mL), and explain what constitutes survival (e.g., all-cause mortality).
Response 3.
We added a more detailed description as suggested for mortality. Viral failure is defined in “2.3. Procedures”.
Comments 4.
Clarify how renal function was measured (e.g., eGFR via CKD-EPI formula) and what time points were used.
Response 4.
Estimated glomerular filtration rate formula used was CKD-EPI as described in “2.5. Statistical Analysis” and this measure was done every six months and the time points analyzed were at 0 and the end of 5th year as described in “2.3. Procedures”.
Comments 5.
Mention statistical methods for survival analysis (e.g., Kaplan–Meier, Cox proportional hazards), adjustments for confounders, and handling of missing data.
Response 5.
This information is described in section "2.5. Statistical Analysis."
Comments 6.
Clarify whether intention-to-treat or per-protocol analysis was used.
Response 6.
In survival analysis both ITT and PP analyses were utilized as described in “3.3 Survival” . In the viral suppression analysis we have clarified that it was ITT analysis.
Comments 7.
Describe the follow-up protocol (e.g., how often patients were monitored, clinical visits, lab tests).
Response 7.
Thank you for the observation. The individuals were evaluated twice a year. We have clarified this in “2.3 Procedures”.
Comments 8.
Include any baseline and follow-up adherence measures if available.
Response 8.
We only checked adherence when there was a viral load above the limit of detection. It is important to notice that the disposal of antiretrovirals is monitored through a national online system.
Comments 9.
Situate the findings in the context of current WHO or national ART guidelines, especially regarding simplification strategies and INSTI-sparing regimens.
Response 9.
As stated in the last paragraph of the discussion section, our findings support the use of bPI plus 3TC or non-nuke as an alternative regimen in special situations.
Comments 10.
Discuss potential reasons for less renal function decline in the simplified arm—e.g., removal of tenofovir or dolutegravir, known for nephrotoxic potential in some patients.
Response 10.
This was discussed in “Discussion” lines 202-210.
Comments 11.
Address the risk of resistance development under bPI-based simplification, especially in settings with limited genotypic resistance testing.
Response 11.
Thank you for the observation. It is our feeling that this is a strength of our study, in the long period of observation there was no viral failure documented with drug resistance mutation. We tried to clarify in the section “3.2 Virological suppression”.
Comments 12.
Mention if genotyping was performed pre-simplification.
Response 12.
Genotyping was not performed prior to ART simplification. It is important to emphasize that all patients were suppressed (at least less than 50 copies/mm³) for at least 1 year prior to being enrolled in the study.
Comments 13.
Acknowledge potential biases (e.g., non-randomized design), small sample size, retrospective components, or missing data.
Response 13.
This was discussed in “Discussion” lines 223-232.
Comments 14.
Note whether cause of death was analyzed or available.
Response 14.
This is a limitation of the study, and might be related to its retrospective and observational nature, which made it impossible to retrieve information in cases of deaths occurring outside of our institution. This can be found in the Discussion section (lines 223-232). We recognize that is one of the limitations of the study.
Comments 15.
Suggest inclusion of newer-generation bPIs or dual regimens (e.g., boosted darunavir + lamivudine) for exploration.
Response 15.
Thank you for your suggestion. We do think there is place for newer drugs in dual regimens, and we add this in our final comments.
Comments on the Quality of English Language: It is recommended that the authors revise the overall English editing and grammar.
The paper has been reviewed by an native English speaker.
Round 2
Reviewer 3 Report
Comments and Suggestions for Authors
The changes are in accordance with the requirements